# DIFFERENTIAL-CRITIC GAN: GENERATING WHAT YOU WANT BY A CUE OF PREFERENCES

## ABSTRACT

This paper proposes Differential-Critic Generative Adversarial Network (DiCGAN) to learn the distribution of user-desired data when only partial instead of the entire dataset possesses the desired properties. Existing approaches select the desired samples first and train regular GANs on the selected samples to derive the user-desired data distribution. DiCGAN introduces a differential critic that can learn the preference direction from the pairwise preferences over the entire dataset. The resultant critic guides the generation of the desired data instead of the whole data. Specifically, apart from the Wasserstein GAN loss, a ranking loss of the pairwise preferences is defined over the critic. It endows the difference of critic values between each pair of samples with the pairwise preference relation. The higher critic value indicates that the sample is preferred by the user. Thus training the generative model for higher critic values encourages the generation of user-preferred samples. Extensive experiments show that our DiCGAN can learn the user-desired data distributions.

## 1 INTRODUCTION

Learning a good generative model for high-dimensional natural signals, such as images (Zhu et al., 2017), video (Vondrick et al., 2016) and audio (Fedus et al., 2018) has long been one of the key milestones of machine learning. Powered by the learning capabilities of deep neural networks, generative adversarial networks (GANs) (Goodfellow et al., 2014) have brought the field closer to attaining this goal. Currently, GANs are applied in a setting where the whole training dataset is of user interest. Therefore, regular GANs no longer meet our requirement when only partial instead of the entire training dataset possesses the desired properties (Killoran et al., 2017). It is more challenging when the given dataset has a small number of desired data.

Adapting vanilla GAN to this setting, a naive way is to first select the samples possessing the desired properties and then perform regular GAN training only on the selected samples to derive the desired distribution. However, vanilla GAN fails when the desired samples are limited. FBGAN overcomes the limited data problem by iteratively introducing desired samples from the generation into the training data. Specifically, FBGAN is pretrained with all training data using the vanilla GAN. In each training epoch, the generator first generates certain amounts of samples. The generated samples possessing the desired properties are selected by an expert selector and used to replace the old training data. Then, regular WGAN is trained with the updated training data. Since the ratio of the desired samples gradually increases in the training data, all training data will be replaced with the desired samples. Finally, FBGAN would derive the desired distribution when convergence. However, bluntly eliminating undesired samples may lead to a biased representation of the real desired data distribution. Because the undesired samples can also reveal useful clues about what is not desired. Suppose we want to generate old face images, however the training data contains only a few old face images whereas it has many young face images. In this case, the young face images can be used as negative sampling (Mikolov et al., 2013) to learn the subtle aging features (e.g. wrinkles, pigmented skin, etc.), which guides the generation of the desired old face images. The conditional variants of GAN, such as CGAN (Mirza and Osindero, 2014) and ACGAN (Odena et al., 2017) can be also applied in this setting by introducing condition variables to model the conditional desired data distribution. However, the generation performance of condition-based GAN is governed by the respective conditions with sufficient training observations. When the desired data is limited, the conditional modeling is dominated by the major classes, i.e., undesired data, resulting in a failure

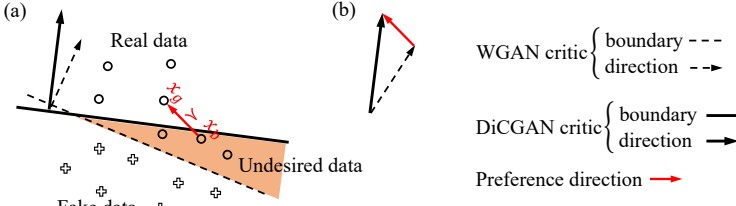

Figure 1: Illustration of why DiCGAN can learn the user-desired data distribution. (a) DiCGAN's critic pushes fake data towards the real desired data while WGAN's critic pushes fake data towards the entire real data. (b) The change of DiCGAN's critic direction is driven by the preference direction. Note that the preference direction is learned from all pairwise preferences.

to capture the desired distribution. All the literature methods require user-defined criteria to select the desired data in order to learn the distribution of the desired data, which may not exist in real applications.

Instead of soliciting a ready-to-use criteria, we consider a more general setting where GAN can be guided towards the distribution of user-desired data by the user preference. In particular, pairwise preferences are the most popular form of user preference due to their simplicity and easy accessibility (Lu and Boutilier, 2011). Therefore, our target is to incorporate pairwise preferences into the learning process of GAN, so as to guide the generation of the desired data.

Relativistic GAN (RGAN) (Jolicoeur-Martineau, 2019) is a variant of regular GAN and is proposed to learn the whole data distribution. It considers the critic value as the indicator of sample quality and defines the discriminator using the difference in the critic values. The critic value in RGAN is similar to the ranking score, but it is used to describe sample quality. Motivated by this, we consider taking the critic value as the ranking score and define the ranking loss for pairwise preferences based on the critic value directly. In particular, the difference in critic values for each pair of samples reflects the user's preference over the samples. This is why we call our critic the **differential critic**, and we propose Differential-Critic GAN (DiCGAN) for learning the user-desired data distribution. As shown in Fig. 1, the differential critic incorporates the user preference direction, which pushes the original critic direction towards the real desired data region instead of the entire real data region. The main contributions are summarized as follows:

- We propose DiCGAN to learn the distributions of the desired data from the entire data using pairwise preferences. To the best of our knowledge, this is the first work to promote the ratio of the desired data by incorporating user preferences directly into the data generation.

- We introduce the differential critic by defining an additional pairwise ranking loss on the WGAN's critic. It endows the difference in the critic values between each pair of samples with user preferences.

- The empirical study shows that DiCGAN learns the distribution of user-desired data and the differential critic can derive the preference direction even from a limited umber of preferences.

## 2 GENERATIVE ADVERSARIAL NETWORKS

Generative Adversarial Network (GAN) (Goodfellow et al., 2014) performs generative modeling by learning a map from low-dimensional input space $\mathcal{Z}$ to data space $\mathcal{X}$, i.e., $G_\theta : \mathcal{Z} \to \mathcal{X}$, given samples from the training data distribution, namely, $x \sim p_r(x)$. The goal is to find $\theta$ which achieves $p_\theta(x) = p_r(x)$, where $p_\theta(x)$ is the fake data distribution $x = G_\theta(z)$. Let $p(z)$ be the input noise distribution and $G$ indicate $G_\theta$. GAN defines a discriminator $D$ that is trained to discriminate real data from fake data to guide the learning of $G$.

Wasserstein GAN (WGAN) (Arjovsky et al., 2017) proposes to use the Wasserstein metric as a critic, which measures the quality of fake data in terms of the distance between the real data distribution and the fake data distribution. The Wasserstein distance (W-distance) is approximated by the difference in the average critic values between the real data and the fake data. The empirical experiments show

that the W-distance between two distributions corresponds well to the quality of the generated data. WGAN's objective function is defined as follows:

$$\min_G \max_D \mathbb{E}_{p_r(x)}[D(x)] - \mathbb{E}_{p_\theta(x)}[D(x)],\tag{1}$$

where $D$ is the critic and satisfies 1-Lipschitz.

## 3 DiCGAN FOR USER-DESIRED DISTRIBUTION

No longer learning the distribution of the whole dataset, GAN is applied in a new scenario, where the distribution of the partial dataset is what we desire. User-desired data may refer to some certain class of data among multiple class datasets, or observations with/without some particular attributes. Such data can be induced from the user preference, which can be represented as an ordering relation between two or more samples in terms of the desired properties. We propose differential-critic GAN to learn the desired data distribution from the user preferences along with the whole dataset.

### 3.1 LEARNING THE DISTRIBUTION OF USER-DESIRED DATA

Following the score-based ranking literature, we suppose that there exists a numeric score associated with each sample, reflecting the user's preference for the sample. A higher score indicates that its corresponding sample is preferred by the user. In detail, let $f$ denote a score function that maps sample $x$ to score $f(x)$. Then, if sample $x$ is desired by the user, its score $f(x)$ exceeds a predefined threshold $\epsilon$, namely, $I(f(x) > \epsilon) = 1$. $I$ is a sign function, which equals 1 if its condition is true and 0 otherwise. For the sake of explanation, we use $p_r(x), p_d(x), p_u(x)$ to denote the distribution of the whole data, the user-desired data and the undesired data, respectively.

FBGAN (Gupta and Zou, 2019) was proposed to learn the distribution of the desired data $p_d(x)$. FB-GAN executes alternatively between two steps: (1) construct the desired dataset $X_d = \{x|I(f(x) > \epsilon) = 1, x \sim p_r(x)\}$; (2) train GAN on $X_d$ to derive $p_d(x)$. However, the assumption that the score function $f$ is predefined in FBGAN may be too restrictive for real applications, where no universal and explicit criteria exists. Further, the definitions of the desired/undesired samples are highly dependent on the choice of the threshold $\epsilon$. The removal of the so-called undesired samples may result in a biased representation of real desired data distribution.

Instead of relying on a predefined score function, we propose to learn the desired data distribution in a straightforward manner from the user preference. Here, we consider a general auxiliary information, i.e., the pairwise preferences, to represent the user preference, due to its simplicity and easy accessibility. For any two samples $x_1, x_2 \sim p_r(x)$, let $x_1 \succ x_2$ denote that $x_1$ is preferred over $x_2$ according to the user-defined criteria. Let X be the training samples, i.e., $X = \{x^i \sim p_r(x)\}$. A collection of pairwise preferences S is obtained by:

$$S = \{s = (x_1, x_2)|x_1 \succ x_2, x_1, x_2 \in X\}.\tag{2}$$

**Definition 1** (Problem Setting). *Given the training samples* X *and the pairwise preferences* S*, the target is to learn a generative model* $p_\theta(x)$ *that is identical to the distribution of the desired data* $p_d(x)$*, i.e.,* $p_\theta(x) = p_d(x)$.

### 3.2 DIFFERENTIAL CRITIC GAN

Instead of WGAN'scritic for quality assessment, we present the differential critic for modelling pairwise preferences. The differential critic can guide the generation of the user-desired data.

#### 3.2.1 PAIRWISE PREFERENCE

In this section, we consider incorporating the pairwise preference into the training of GAN.

The score-based ranking model (Zhou et al., 2008) is used to model the pairwise preferences. It learns the score function $f$, of which the score value, called ranking score in the model, is the indicator of the user preference. Further, the difference of ranking scores can indicate the pairwise preference relation. That is, for any pair of samples $x_1, x_2$, if $x_1 \succ x_2$ then $f(x_1) - f(x_2) > 0$ and vice versa. For any pairwise preference $s : x_1 \succ x_2$, the ranking loss we consider is as follows:

$$h(s) = \max\left(0, -\left(f(x_1) - f(x_2)\right) + m\right),\tag{3}$$

where $m$ is the ranking margin. For other forms of ranking losses, the reader can refer to (Zhou et al., 2008).

Instead of learning the score function independent of GAN's training, we consider incorporating it into GAN's training, guiding GAN towards the generation of the desired data. The critic in RGAN (Jolicoeur-Martineau, 2019) is similar to the score function, where the critic values are used to describe the sample quality. We are motivated to take the critic value as the ranking score and define the ranking loss on the critic value directly. In particular, the difference in the critic values for each pair of samples reflects the user's preference over the samples.

**Remark 1** (Pairwise regularization to the generator). *It is possible to consider a pairwise regularization to the generator. As the target is to learn the desired distribution, the regularization to the generator can be used to make the critic values of the generated samples larger than those of the undesired samples. We construct the regularization with the principle similar as FBGAN. Specifically, a selector is first applied to give a full ranking for the training data and then bottom $K$ samples are picked up as the undesired samples. The pairwise preferences are then defined over the generated samples and the undesired samples.*

### 3.2.2 LOSS FUNCTION

We build DiCGAN based on WGAN and the pairwise ranking loss is defined over the WGAN's critic. The loss function for DiCGAN is defined as:

$$\min_G \max_D \mathbb{E}_{p_r(x)}\left[D(x)\right] - \mathbb{E}_{p_\theta(x)}\left[D\left(x\right)\right] - \lambda\frac{1}{|\mathrm{S}|}\sum_{s\in\mathrm{S}}\left[h\left(s\right)\right]. \tag{4}$$

where $h(s)$ is the pairwise ranking loss (equation 3). $\lambda$ is a balance factor, which will be discussed further in section 3.3. Similar to WGAN, we formulate the objective for the differential critic $L_D$ and the generator $L_G$ as:

$$L_D = \frac{1}{b}\sum_{i=1}^{b} D(x^i) - D(G(z^i)) - \lambda\frac{1}{n_\mathrm{s}}\sum_{j=1}^{n_\mathrm{s}} h(s^j), \qquad L_G = \frac{1}{b}\sum_{i=1}^{b} -D(G(z^i)), \tag{5}$$

where $b$ is the batch size. This is the same for the fake samples. $n_\mathrm{s}$ is the number of pairs sampling from S.

The advantages of DiCGAN are twofold: (1) The introduced ranking loss in DiCGAN is defined on the critic value. Apart from WGAN, it can be easily applied to other GAN variants developed based on the critic, e.g., RGAN. (2) DiCGAN leverages the entire dataset. The pairwise preferences are constructed on the whole dataset. Thus the undesired samples are also utilized during the training.

In the following, we argue that the differential critic in DiCGAN can guide the generator to learn the user-desired distribution from two aspects. (1) As shown in Fig. 1, the differential critic in DiCGAN provides the direction towards the real desired data. We denote the critic direction as the moving direction of fake data, which is orthogonal to the decision boundary of the critic. Referring to equation 4, DiCGAN's critic loss consists of two terms: the vanilla WGAN loss and the ranking loss. The vanilla WGAN loss imposes the critic direction from the fake data to the real data. Meanwhile, the ranking loss induces a user preference direction, which points from the undesired data to the desired data. Combining these two effects, the critic direction of DiCGAN targets for the region of the real desired data only. (2) DiCGAN assigns high critic values for user-desired data and promotes the generation of samples with high critic values. The vanilla WGAN loss encourages the critic to assign high critic values for real data and low critic values for fake data. Meanwhile, the ranking loss encourages high critic values to be assigned to the real desired data while low critic values are assigned to real undesired data. Therefore, the real desired data achieves high critic values from the critic. Similar to WGAN, the training paradigm of DiCGAN promotes the generation of samples with high critic values, which is equivalent to encouraging the generation of user-desired data in DiCGAN.

### 3.3 REFORMULATING DiCGAN TO ENSURE DATA QUALITY

Let us revisit the objective of DiCGAN (equation 4). The first two terms of equation 4 can be considered as the WGAN regularisation, which ensures the generated data distribution is close to the

---

**Algorithm 1** Training algorithm of DiCGAN

1: **input:** training data X, pairwise preferences S
2: **initilization:** balance factor $\lambda$, #generated samples $n_{\mathrm{g}}$, #pairs $n_{\mathrm{s}}$, batchsize $b$, #iterations per epoch $n_{\mathrm{i}}$, #critic iterations per generator iteration $n_{\mathrm{critic}}$
3: Pretrain $D$ and $G$
4: **repeat**
5:    % Shift to the user-preferred distribution
6:      Generate samples using equation 7
7:      Replace old samples in X with $X_{\mathrm{g}}$ using equation 8
8:    Obtain pairwise preferences R using equation 2
9:    % Training of $D$ and $G$ at an epoch
10:    **for** $i = 1, \ldots, n_{\mathrm{i}}$ **do**
11:      **for** $t = 1, \ldots, n_{\mathrm{critic}}$ **do**
12:        Sample $\{x^i\}_{i=1}^b$ from X, $\{z^i \sim p(z)\}_{i=1}^b$
13:        Sample $\{s^j\}_{j=1}^{n_{\mathrm{s}}}$ from S.
14:        Train the differential critic $D$ using $L_D$ in equation 5
15:      **end for**
16:      Train the generator $G$ using $L_G$ in equation 5
17:    **end for**
18: **until** converge

---

whole real data distribution, i.e., $p_\theta \approx p_{\mathrm{r}}$. The third term serves as a correction of WGAN, which makes WGAN slightly biased to our target of learning the desired data distribution, i.e., $p_\theta = p_{\mathrm{d}}$.

Therefore, the WGAN regularisation serves as the cornerstone of our DiCGAN. Particularly, if the desired data distribution is close to the entire data distribution, the rank loss easily corrects the WGAN to achieve the desired data distribution. Otherwise, the satisfactory performance of DiCGAN may require the online hyperparameter tuning of $\lambda$ during the training process.

According to the above analysis, we consider reformulating the objective of DiCGAN, i.e., equation 4 into an equivalent objective with a hard WGAN constraint:

$$\min_G \max_D -\sum_{s \in \mathrm{S}} [h(s)], \quad \text{s.t.} \ d(p_{\mathrm{r}}, p_\theta) = \left| \mathbb{E}_{p_{\mathrm{r}}(x)} [D(x)] - \mathbb{E}_{p_\theta(x)} [D(x)] \right| \leq \varepsilon. \tag{6}$$

where $\varepsilon > 0$. Note that we impose an explicit non-negative constraint on $d(p_{\mathrm{r}}, p_\theta)$, to highlight that it is a distance metric. It is still equivalent to WGAN loss from its definition. Therefore, equation 4 is the Lagrangian function. Since equation 6 imposes a hard constraint on the WGAN loss, it is more difficult to optimize compared to equation 4. However, more efficient solutions of DiCGAN can be explored by analyzing equation 6 regarding the hard constraint on $d(p_{\mathrm{r}}, p_\theta)$.

In terms of a minor correction situation, this means the desired data distribution $p_{\mathrm{d}}$ is close to the real data distribution $p_{\mathrm{r}}$. Therefore, the hard constraint dominates the training goal of DiCGAN. By assigning a proper $\lambda$ to ensure the constraint is satisfied, equation 4 can learn the distribution of the user-desired data while ensuring data quality.

In terms of a major correction situation, this means the desired data distribution $p_{\mathrm{d}}$ is quite diverse from the real data distribution $p_{\mathrm{r}}$. Therefore, DiCGAN needs to achieve an equilibrium between the correction, imposed by the ranking loss, and the hard constraint, imposed by the WGAN loss. However, a large correction may not ensure the quality of the generated data, since the WGAN loss, used to guarantee the image quality, is defined between the generated data and the whole real data. To avoid the major correction, we propose to break the major correction into a sequence of minor corrections to ensure data quality. Namely, at each epoch, we first use the generator $G$ to generate $n_{\mathrm{g}}$ samples, denoted as $X_{\mathrm{g}}$:

$$X_{\mathrm{g}}^e \leftarrow \{G^e(z^1), \ldots, G^e(z^{n_{\mathrm{g}}})\}, \quad \{z^i \sim p(z)\}_{i=1}^{n_{\mathrm{g}}}, \tag{7}$$

where $e$ is the epoch index. Then we replace the old training samples with the generated samples:

$$X^{e+1} \leftarrow X^e \setminus X_{\mathrm{o}}^e \cup X_{\mathrm{g}}^e, \tag{8}$$

where $X_{\mathrm{o}}^e$ are the old (least-recently added) $n_{\mathrm{g}}$ samples in $X^e$.

Due to the ranking loss, the generated data distribution $p_\theta^e$ is closer to the desired data distribution $p_{\mathrm{d}}$, compared to the constructed $p_{\mathrm{r}}^e$ at each epoch. Therefore, the iterative replacement (equation 8)

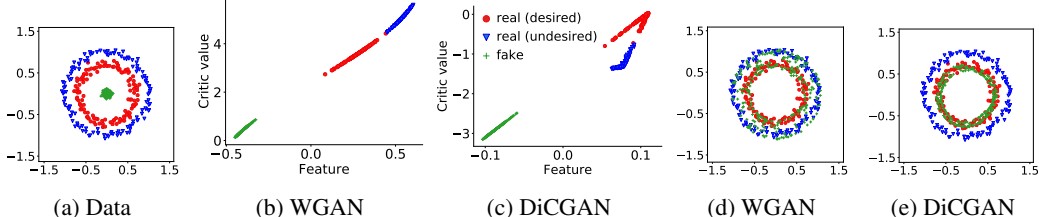

| (a) Data | (b) WGAN | (c) DiCGAN | (d) WGAN | (e) DiCGAN |

Figure 2: (a-c) Comparison of the critic in DiCGAN and WGAN. DiCGAN's critic can assign higher critic values for real desired data than real undesired data while WGAN's critic cannot. (d-e) The visualization of the generated samples from DiCGAN and WGAN. Learning the desired distribution means that the fake data should overlap with the real desired data only.

can gradually shift the real data distribution $p_{\mathrm{r}}$ towards the desired data distribution $p_{\mathrm{d}}$. Namely, $d(p_{\mathrm{r}}, p_{\mathrm{d}}) > \cdots > d(p_{\mathrm{r}}^e, p_{\mathrm{d}}) > d(p_{\mathrm{r}}^{e+1}, p_{\mathrm{d}}) > \cdots$. So that only a minor correction needs to be imposed on $p_{\theta}^e$ through optimizing equation 4 at each epoch. Iteratively, the generated distribution $p_{\theta}$ shifts towards $p_{\mathrm{d}}$. The training algorithm is summarized in Algorithm 1. For the sake of easy optimization, we pretrain the differential critic $D$ and the generator $G$ using WGAN.

## 4 CASE STUDY ON SYNTHETIC DATA

To gain an intuitive understanding of the difference between our DiCGAN and WGAN regarding the critic and the generator, we conduct a case study on the synthetic dataset.

The synthetic dataset consists of two concentric circles by adding Gaussian noise with a standard deviation of 0.05 (See Fig. 2a). The samples located on the inner circle are considered to be the desired data, while the samples on the outer circle are defined as the undesired data. By labeling the desired data as $y = 1$ and the undesired data as $y = 0$, we can construct the pairwise preference for two samples $x_1$ and $x_2$ based on their labels. Namely $x_1 \succ x_2$ if $y_1 = 1 \wedge y_2 = 0$, and vice versa. The pairs are constructed within each mini-batch. Our target is to learn the distribution of the desired data (i.e., samples on the inner circle), using the whole data along with the constructed pairwise preferences.

### 4.1 WGAN VS DICGAN ON CRITIC

Experiment setting: we fix the generator and simulate the fake data as the 2D Gaussian blob with a standard deviation of 0.05 (green pluses). We first train the critic to converge. Then, we project the output on the second last layer of the critic into 1D space using kernel principal components analysis (Schölkopf et al., 1997), to derive the projected features. To explore the difference between the critics of WGAN and DiCGAN, we draw the curve of the critic values versus the projected features for WGAN and DiCGAN, respectively (Fig. 2b, 2c).

From Fig. 2b, 2c, we can see: $(1)$ in terms of the real data and the fake data, the critic of both WGAN and DiCGAN can achieve perfect discrimination. Meanwhile, the projected features of the real data and those of the fake data are also completely separated; $(2)$ in terms of the real desired data and the real undesired data, the critic of DiCGAN assigns higher values to the desired samples, compared to the undesired samples. This is because our ranking loss expects a higher ranking score (i.e., critic values) for the corresponding desired data. $(3)$ In contrast, the critic of WGAN assigns lower values to the desired data, since the desired data is closer to the fake data compared to the undesired data.

### 4.2 WGAN VS DICGAN ON GENERATOR

Experiment setting: we train the critic and the generator following the regular GANs' training procedure. The generation results of WGAN and DiCGAN is shown in Fig. 2d, 2e.

DiCGAN (shown in Fig. 2e) only generates the user-desired data, i.e., generated data covering the inner circles, while WGAN (shown in Fig. 2d) generates all data, i.e., generated data covering the inner and outer circles. As the critic in DiCGAN can guide the fake data towards the real data region and away from the undesired data region, the generator thus produces data which is similar to the real desired data. Because the critic in WGAN pushes the fake data to the real data region only, the generator finally produces whole real-alike data.

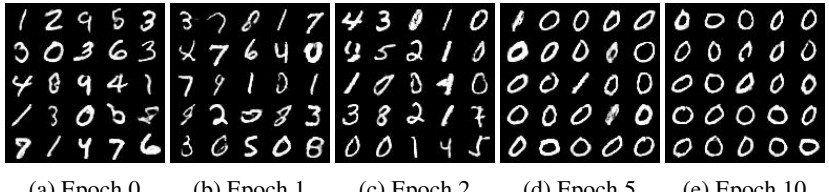

| (a) Epoch 0 | (b) Epoch 1 | (c) Epoch 2 | (d) Epoch 5 | (e) Epoch 10 |
|---|---|---|---|---|

Figure 3: Generated digits of DiCGAN on MNIST. DiCGAN learns the distribution of small digits.

## 5 EXPERIMENTAL STUDY

DiCGAN is applied to learn the distribution of the desired data on MNIST (Lecun et al., 1998) and CelebA-HQ (Karras et al., 2018) datasets. Due to the limited space, we present more experiment results in the appendix.

**Networks & Hyperparameters** The balance factor $\lambda$ and the ranking margin $m$ is set to $1$. We adopt the same network structures as in (Gulrajani et al., 2017). See the appendix for other settings.

**Evaluation Metric** To evaluate the performance of learning the desired data distribution, we calculate the percentage of user-desired data in GAN's generation, i.e., the ratio of the desired generated data to the whole generated data (D/W). We use inception score (IS) (Salimans et al., 2016) and multiscale structural similarity (MS-SSIM) (Odena et al., 2017) to evaluate the quality and intra-class diversity for GANs' generation, respectively.

**Baselines** We compare DiCGAN with WGAN (Gulrajani et al., 2017), CWGAN (Mirza and Osindero, 2014) and FBGAN (Gupta and Zou, 2019). WGAN is only trained with the desired data to derive the desired distribution. CWGAN is the extension of GAN with a conditional label $c$. To train CWGAN, we split the training data into the desired class ($c = 1$) and the undesired class based on a predefined user criteria ($c = 0$). Then $p(x|c = 1)$ is the desired data distribution. FBGAN adopts an iterative training paradigm to derive the desired data distribution. At each training epoch, FBGAN resorts to an extra selector to select the desired samples from the generated samples and performs regular GAN training using the selected desired samples. ACGAN (Odena et al., 2017) also shows poor results like CWGAN when the desired data is limited, so we do not report here.

### 5.1 LEARNING THE DISTRIBUTION OF DESIRED DIGITS

We design the experiment to learn the distribution of small digits in MNIST. We use $50K$ $28 \times 28$ images as the training data. Zero is the smallest digit in MNIST, thus as the desired data,

As for WGAN and CWGAN, zero digits in the training data are regarded as the desired samples ($c = 1$), whose size is $4,950$. The other digits are labeled as the undesired samples, whose size is $45,050$ ($c = 0$). WGAN is only trained with the constructed desired data. CWGAN conditions on $c$ to model a conditional data distribution $p(x|c)$ for MNIST dataset. FBGAN and our DiCGAN resort to a classifier, pre-trained for digit classification, to obtain the labels for the generated samples. At every training epoch, FBGAN generates 50000 samples and requests the classifier to label them. Then the images are ranked using the predicted label, with the smaller digit ranked higher. The generated images with digits ranked in the top $50\%$, i.e., small digits, are selected as the desired data to replace old training data. As for DiCGAN, the pairwise comparison can be obtained for two images $x_1$ and $x_2$ according to their predicated label $y_1$ and $y_2$, namely $x_1 \succ x_2$ if $y_1 < y_2$, and vice versa. At each iteration, we construct $32$ pairwise preferences for each mini-batch with $64$ training samples.

Fig. 3 presents the generated MNIST images randomly sampled from the generator of DiCGAN. It shows that the generated MNIST digits gradually shift to smaller digits during the training, and converge to the digit zero. We sample $50K$ samples from the generators of various GANs and respectively calculate the percentage of digit zero and digits zero to four among the generated digits for a quantitative evaluation. In Table 1, only small digits are generated by DiCGAN and FBGAN; WGAN and CWGAN can also learn the distribution of the desired digit since the dataset is simple and has relatively sufficient data for the desired digit. However, WGAN and CWGAN do not exhibit a smooth convergence to digit zero like FBGAN and DiCGAN (See Fig. 10 in the appendix). In addition, when the dataset is complex and the desired data is insufficient, WGAN and CWGAN fail, which is described in the next section.

### 5.1.1 COMPARISON OF DICGAN AND FBGAN

Though FBGAN achieves good performance in learning the desired data distribution, it requires a lot of supervision information from the selector. We calculate the number of effective pairs (#EP) used in DiCGAN and FBGAN, respectively. #EP in DiCGAN denotes the total number of explicitly constructed pairs during the training, i.e., #EP $= \sum_{i=1}^{n_e} \sum_{j=1}^{n_i} n_s$. FBGAN selects the desired samples from the generated samples. #EP can be induced by the implicit pairs implied by the desired samples versus the undesired samples, i.e., #EP $= \sum_{i=1}^{n_e} n_{gd} \times n_{gu}$, where $n_e$ is the number of training epochs. where $n_{gd}$ and $n_{gu}$ denote the number of desired samples and undesired samples in the generation, respectively.

Fig. 4a shows that (1) the #EP used in DiCGAN is much smaller than that in FBGAN at each training epoch; (2) the total #EP used in DiCGAN is significantly less than that in FBGAN, which can be reflected from the shadow area. In total, DiCGAN used $9.53e4$ effective pairs while FBGAN used $2.02e8$ effective pairs. Our DiCGAN is scalable to the large training dataset, e.g. MNIST. #EP in DiCGAN is linearly correlated to the training size. In contrast, #EP in FBGAN is determined by $n_{gd}$ and $n_{gu}$, which are both linearly correlated to the training size. Thus #EP in FBGAN thus is quadratically correlated to the training size.

### 5.1.2 COMPARING DICGAN AND FBGAN GIVEN THE LIMITED SUPERVISION

We compare DiCGAN and FBGAN on MNIST dataset given the limited supervision. Specifically, the query amount of resorting to the pre-trained classifier to obtain the prediction of the generated samples is restricted to $5,000$ for both FBGAN and DiCGAN.

Fig. 4d shows that DiCGAN can learn the desired distribution while FBGAN fails, only generating $10.3\%$ digit zero, which is consistent with the visual results in Fig. 4b and Fig. 4c. This supports the claim that the negative samples are beneficial to learn the user-desired distribution. Particularly, the preference direction can be captured by our differential critic even the supervision is limited, which guides the generation towards the desired data.

### 5.2 LEARNING THE DISTRIBUTION OF FACE WITH THE DESIRED ATTRIBUTE

We consider old face images as the desired data and design the experiment to learn the distribution of old face images on CelebA-HQ. CelebA-HQ dataset is the high quality version of a subset from Celeb Faces Attributes (CelebA) dataset, which consists of 30K images face images of celebrities, annotated with $40$ binary attributes such as age. We resize the images to $64 \times 64$. The training setting for each GAN is similar to those mentioned above. (See the appendix for more details).

In Fig. 5, we visualize the generated face images randomly sampled from the generator of each model. (1) WGAN has poor generation since its training data, i.e., the desired subset is insufficient. (2) CWGAN has good quality of generation but fails to capture the desired distribution. There is only two old face image out of 9 randomly sampled images. (3) All sampled images from FBGAN and our DiCGAN are the desired old faces. We sample $10K$ samples from the generator and calculate the percentage of old

| Method | MNIST D/W | | CelebA-HQ | | |
|---|---|---|---|---|---|
| | top 1 | top 5 | D/W | IS | MS-SSIM |
| Original | 9.9 | 51.1 | 22.1 | **2.81** | **0.52** |
| WGAN (subset) | 97.3 | 98.2 | 73.8 | 1.77 | 0.60 |
| CWGAN | 95.0 | 96.4 | 16.4 | 2.12 | 0.65 |
| FBGAN | **100.0** | **100.0** | 98.2 | 2.05 | 0.60 |
| DiCGAN | **100.0** | **100.0** | 99.5 | 1.93 | 0.60 |

Table 1: Results on the MNIST and CelebA-HQ dataset. Top $1$ refers to digit zero. Top $5$ refers to digits zero to four.

face images among the generated samples for quantitative evaluation. In Table 1, almost all the images generated by DiCGAN and FBGAN are old face images. WGAN and CWGAN both fail to

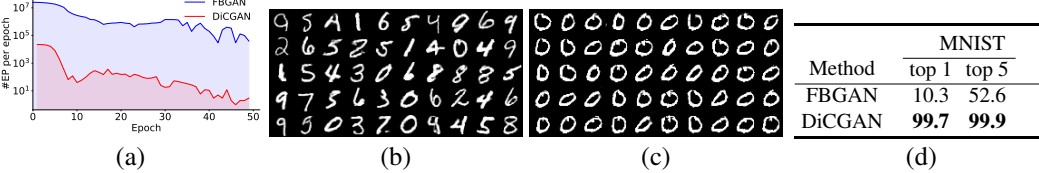

(a)  (b)  (c)  (d)

| Method | MNIST | |
|---|---|---|
| | top 1 | top 5 |
| FBGAN | 10.3 | 52.6 |
| DiCGAN | **99.7** | **99.9** |

Figure 4: (a) Comparison between DiCGAN and FBGAN w.r.t. #EP per epoch on MNIST in Section 5.1. (b) The generated results of FBGAN in Section 5.1.2. (c) The generated results of FBGAN in Section 5.1.2. (d) % of desired digits (D/W) of FBGAN and DiCGAN in Section 5.1.2.

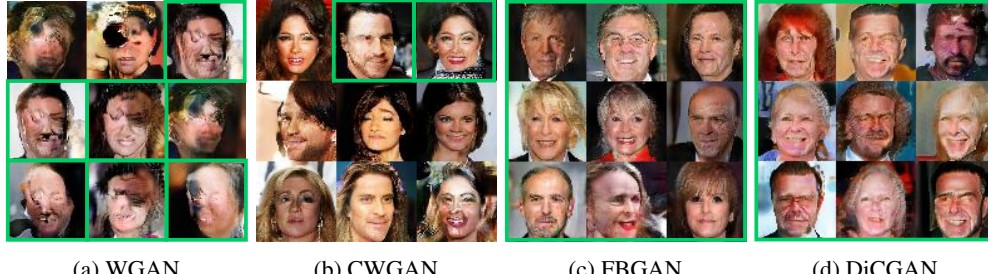

| (a) WGAN | (b) CWGAN | (c) FBGAN | (d) DiCGAN |

Figure 5: Generated images by WGAN, CWGAN, FBGAN and DiCGAN. The green boxes refer to the images which are classified as old images by a trained classifier.

capture the distribution of old face images. As for the quality and the diversity, (1) GAN shows the lowest IS, meaning a poor-quality of generation. The generations of other GANs except for WGAN present similar IS and MS-SSIM. (2) We calculate the IS of the old faces in the training data (denoted as "original" in Table 1. The IS and MS-SSIM of the generations do not exhibit a big difference from the "original", which means that the quality and the diversity of the generations are relatively good.

## 6 DISCUSSIONS

Current GAN (Goodfellow et al., 2014; Mirza and Osindero, 2014; Odena et al., 2017; Gupta and Zou, 2019) based methods require user-defined criteria to select the desired data in order to derive the distribution of the desired data. There are two limitations for these methods: (1) the criteria are not always accessible in real applications; (2) eliminating the handcrafted undesired samples loses useful information about what is not desired. Other works (Gómez-Bombarelli et al., 2018; Engel et al., 2018) proposed to find the subset of the latent space corresponding to the desired data and generates data from the latent subset. However, they also rely on user-defined criteria for labeling the desired data or a ready-to-use score function to find the subset of the latent space with high scores.

This paper proposes DiCGAN to learn the distributions of the user-desired data from the entire data using the pairwise preferences. We empirically demonstrate the efficacy of DiCGAN in terms of promoting samples with user-desired properties on MNIST and CelebA-HQ datasets, respectively. One promising future direction for DiCGAN could be the minority population promotion for imbalanced data tasks, such as imbalanced classification problems, few-shot learning, one-shot learning or even an open-set problem. Another interesting direction of DiCGAN could be the desired policy generation in imitation learning if the pairwise comparison between the policies can be properly designed.

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

## A    COMPARISON OF DICGAN AND FBGAN

We plot the ratio of generated zero digit to the whole generated data (D/W) of DiCGAN and FBGAN during the training process in Fig. 6a. It shows that DiCGAN converges faster than FBGAN.

We explore the gap in the performance between DiCGAN and FBGAN evolves as the number of supervision increases on MNIST. Specifically,the query amount of resorting to the pre-trained classifier to obtain the prediction of the generated samples is restricted to $5K, 50K, 100K, 150K, 180K, 200K, 500K$ for both FBGAN and DiCGAN.

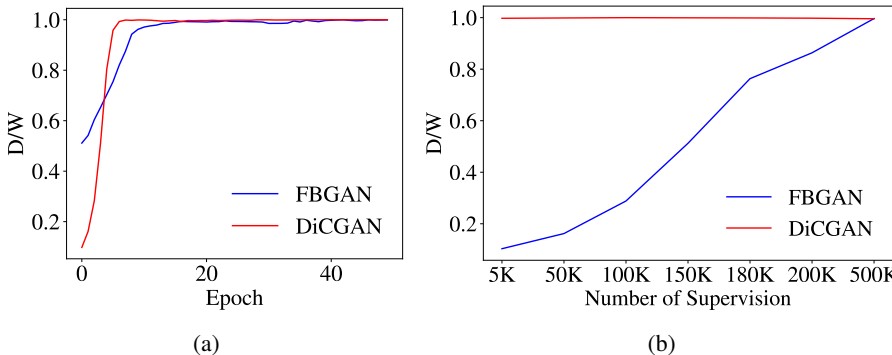

(a)                                    (b)

Figure 6: Comparison of DiCGAN and FBGAN. (a) plots the ratio of generated zero digit to the whole generated data (D/W) versus the training epoch for FBGAN and DiCGAN, respectively. (b) plots the ratio of generated zero digit to the whole generated data (D/W) versus the number of supervision for FBGAN and DiCGAN, respectively.

Fig. 6b plots D/W versus the number of supervision for FBGAN and DiCGAN, respectively. It shows that (1) DiCGAN always learns the desired data distribution even given the limited supervision; (2) when given the limited supervision, FBGAN fails to learn the desired distribution, i.e., achieving a small D/W; (3) FBGAN performs better and achieves a higher D/W, narrowing the performance gap with DiCGAN as the number of supervision increases.

## B    ABLATION STUDY

The objective in our DiCGAN (equation 4) consists of two components, i.e., the WGAN loss, which serves as the cornerstone of DiCGAN, and the ranking loss, which serves as the correction for WGAN. Meanwhile, we introduce the operation of replacement (equation 8) during the model training.

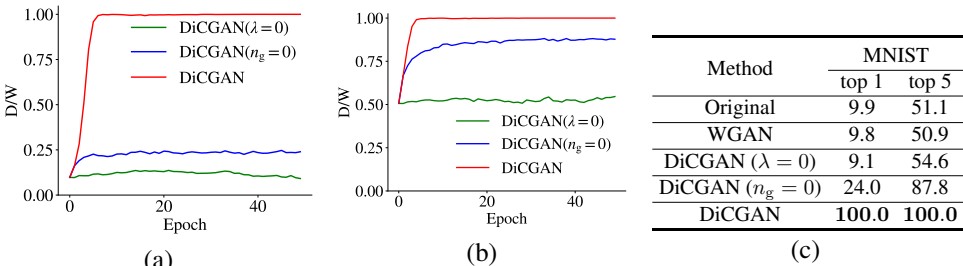

| Method | MNIST | |
|---|---|---|
| | top 1 | top 5 |
| Original | 9.9 | 51.1 |
| WGAN | 9.8 | 50.9 |
| DiCGAN ($\lambda = 0$) | 9.1 | 54.6 |
| DiCGAN ($n_g = 0$) | 24.0 | 87.8 |
| DiCGAN | **100.0** | **100.0** |

(a)                          (b)                          (c)

Figure 7: (a-b) The percentage of desired samples (D/W) versus epoch in DiCGAN ($\lambda = 0$), DiCGAN ($n_g = 0$) and DiCGAN. (a) plots the D/W of the digit zero. (b) plots the D/W of the digit zero to four. (c) The percentage of desired samples (D/W)from the orginial datasets, WGAN, DiCGAN ($\lambda = 0$), DiCGAN ($n_g = 0$) and DiCGAN.

To analyze the effects of the correction for WGAN (the third term in equation 5) and the replacement operation, we plot the percentage of desired samples (D/W) versus the training epoch for DiCGAN ($\lambda = 0$), DiCGAN ($n_g = 0$) and DiCGAN in Fig. 7a, 7b. Meanwhile, the the converged percentage of desired samples (D/W) are reported in Fig. 7c. It can be seen that

1. **Without the correction term ($\lambda = 0$), DiCGAN cannot learn the desired data distribution.** The percentage of desired samples (D/W) from DiCGAN ($\lambda = 0$) remains constant during training on MNIST (Fig. 7a, 7b) compared with the original datasets (Fig. 7c). This is because that the remaining WGAN term in DiCGAN($\lambda = 0$) focuses on learning the training data distribution.

2. **Without the replacement ($n_g = 0$), DiCGAN makes a minor correction to the generated distribution.** In Fig. 7a, 7b, the D/W of DiCGAN ($n_g = 0$) slightly increases compared with

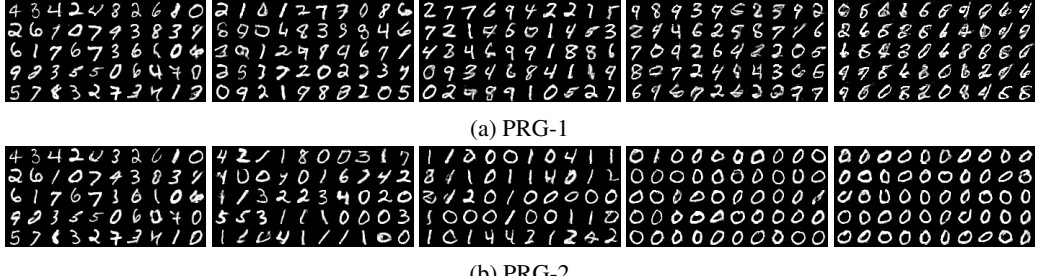

(a) PRG-1

(b) PRG-2

Figure 8: Generated digits of PRG-1 and PRG-2 on MNIST during the training process.

the original datasets. This is consistent with our analysis that the correction term would drive the generation towards the desired distribution.

3. **DiCGAN learns the desired data distribution with a sequential minor correction.** The D/W of DiCGAN grows with training and reaches almost $100\%$ when convergence. The correction term drives DiCGAN's generation towards the desired data slightly at each epoch. With the iterative replacement, the minor correction sequentially accumulates and finally the generated distribution shifts to the desired data distribution.

## C  PAIRWISE REGULARIZATION ON THE GENERATOR

As discussed in Remark 1, the pairwise regularization is possibly added to the generator. We consider two cases of adding the regularization to the generator. First, we only add the pairwise regularization to the generator (PRG-1). Second, we add the regularization to the generator together with the regularization on the critic (PRG-2).

The objective for PRG-1 is as follows:

$$L_D = \mathbb{E}_{p_r(x)}\left[D(x)\right] - \mathbb{E}_{p_\theta(x)}\left[D(x)\right], \tag{9}$$

$$L_G = \mathbb{E}_{p_\theta(x)}\left[D(x)\right] - \lambda_g \frac{1}{|\mathrm{S}'|} \sum_{s \in \mathrm{S}'} \left[h(s)\right]. \tag{10}$$

where $h(s)$ is equation 3. $\mathrm{S}'$ is the pairwise preferences constructed between the generated data and the undesired data, i.e., $\mathrm{S}' = \left\{s = (x_1, x_2) | x_1 \succ x_2, x_1 \sim p_\theta(x), x_2 \sim p_u(x)\right\}$. Now the generator consists of two terms, the original WGAN loss on the generator aims to achieve $\mathbb{E}_{p_\theta(x)}\left[D(x)\right] > \mathbb{E}_{p_r(x)}\left[D(x)\right]$, while the regularization aims to achieve $\mathbb{E}_{p_\theta(x)}\left[D(x)\right] > \mathbb{E}_{p_u(x)}\left[D(x)\right]$. Since the undesired data is subset of the real data, i.e., $\{x | x \sim p_u(x)\} \subseteq \{x | x \sim p_r(x)\}$, the WGAN loss always dominates the training of the generator. Therefore, PRG-1 degenerates to WGAN.

The objective for PRG-2 is as follows:

$$L_D = \mathbb{E}_{p_r(x)}\left[D(x)\right] - \mathbb{E}_{p_\theta(x)}\left[D(x)\right] - \lambda \frac{1}{|\mathrm{S}|} \sum_{s \in \mathrm{S}} \left[h(s)\right], \tag{11}$$

$$L_G = \mathbb{E}_{p_\theta(x)}\left[D(x)\right] - \lambda_g \frac{1}{|\mathrm{S}'|} \sum_{s \in \mathrm{S}'} \left[h(s)\right], \tag{12}$$

where $\mathrm{S}$ is constructed based on equation 2. Although the generator consists of two terms, same to our analysis about PRG-1, the extra pairwise regularization on the generator is invalid. Meanwhile, the extra pairwise regularization on the critic works like that in DiCGAN. Therefore, the whole framework degenerates to DiCGAN.

We conducted the experiments on MNIST to show the effectiveness of these two methods. $\lambda$ and $\lambda_g$ are both set to 1. Fig. 8 shows the generated digits of PRG-1 and PRG-2 during the training process. PRG-1 failed to learning the desired distribution. PRG-2 can learn the desired distribution. The quantitative results are consistent with the visual results, with $13.9\%$ and $99.4\%$ D/W, respectively.

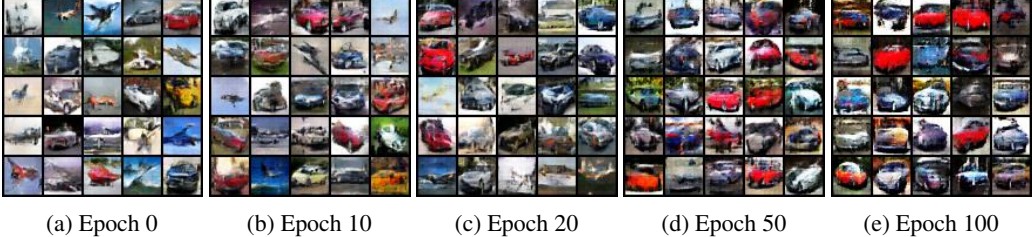

| (a) Epoch 0 | (b) Epoch 10 | (c) Epoch 20 | (d) Epoch 50 | (e) Epoch 100 |

Figure 9: Generated images of DiCGAN on CIFAR. DiCGAN aims to learn the distribution of car images of CIFAR. The training dataset is composed of car and plane images in CIFAR-10.

## D   LEARNING THE DISTRIBUTION OF DESIRED OBJECTS

We consider cars as the desired objects and design the experiment to learn the distribution of cars in CIFAR.

**Sample selection in FBGAN:** A classifier, pretrained for classifying cars and planes, is adopted for selection. The generated objects, classified to car, are selected to replace the old training data.

**Pairwise preferences construction in DiCGAN:** Denoting label of CIFAR image as $y$, the pairwise preference between two images $x_1$ and $x_2$ are $x_1 \succ x_2$ when $y_1 = $ "$car$", $y_2 = $ "$plane$", and vice versa. At each iteration, we construct 32 pairs by random sampling pairs from the mini-batch 64 samples.

In Fig. 9, we visualize the generated CIFAR images randomly sampled from the generator of DiCGAN. It shows that DiCGAN gradually generates cars, as we desired.

Meanwhile, we sample $10K$ samples from the generator and calculate the percentage of car images among the generated samples for quantitative evaluation. In Table 2, (1) almost all images generated by DiCGAN and FBGAN are car images; (2) the percentage of car images generated by WGAN is similar to the training dataset.

| Method | CIFAR (IS) | | |
|--------|------|------|---------|
|        | D/W  | IS   | MS-SSIM |
| Dataset | 50.0 | **4.96** | **0.45** |
| FBGAN  | 93.9 | 3.78 | 0.52 |
| DiCGAN | **95.4** | 3.67 | 0.49 |

Table 2: Results on CIFAR dataset.

## E   EXPERIMENT SETTINGS

**Hyperparameter** The batch size $b$ is set to 50 for MNIST, 64 for CIFAR and CelebA-HQ datasets. The #generated samples $n_g$ is set to $50K$ for MNIST, $1K$ for CIFAR and $3,000$ for CelebA-HQ, respectively. Other hyperparameters are adopted the same as in Gulrajani et al. (2017).

We construct pairwise preferences using the minibatch samples at each iteration based on the classification labels. We construct the pairs by randomly selecting two samples from the minibatch samples, respectively. The pairs, in which two samples belong to the same class, i.e., same digits or same objects, are removed.

**CelebA-HQ training setting** WGAN is only trained with the constructed desired dataset. CWGAN conditions on $c$ to model a conditional data distribution $p(x|c)$. There are $6,632$ samples labeled as desired and $23,368$ samples labeled as undesired in the training data. A classifier, pre-trained for classifying young faces and old faces, is adopted for predicting the labels for the generated face images. At every training epoch, FBGAN generates $3,000$ images and those classified as the old are selected to replace the old training data. As for DiCGAN, the generated face image classified with the old attribute is preferred over the face image classified with the young attribute. At each iteration, we construct 32 pairs by random sampling pairs from the mini-batch 64 samples.

## F   MORE VISUAL RESULTS

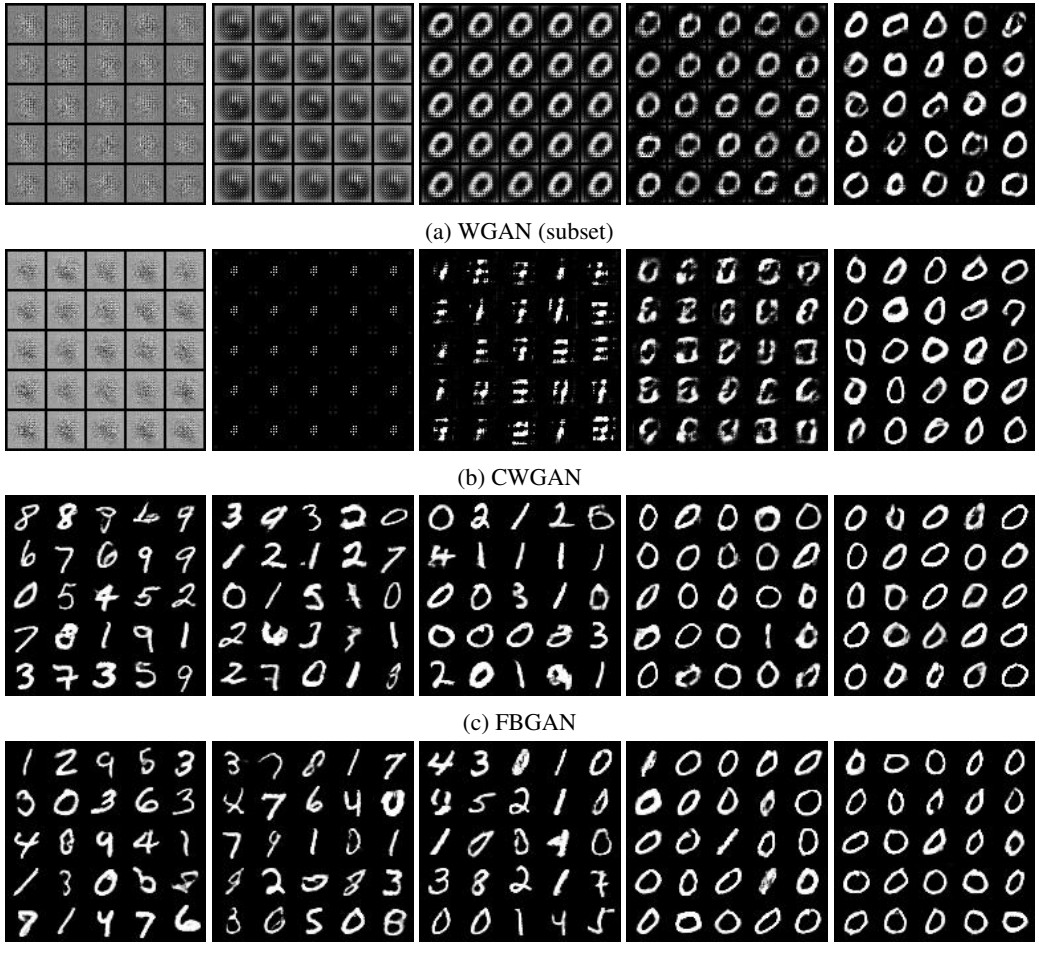

(a) WGAN (subset)

(b) CWGAN

(c) FBGAN

(d) DiCGAN

Figure 10: Generated digits of DiCGAN on MNIST during the training process.

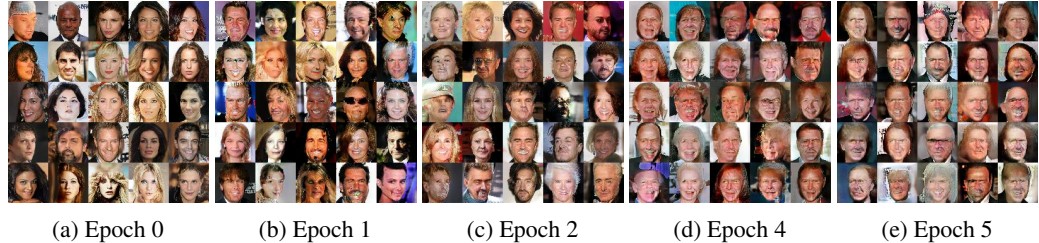

(a) Epoch 0     (b) Epoch 1     (c) Epoch 2     (d) Epoch 4     (e) Epoch 5

Figure 11: Generated images of DiCGAN on CelebA-HQ. DiCGAN learns the distribution of old faces. DiCGAN gradually generates more old face images.

