# OpenReview forum: "Differential-Critic GAN: Generating What You Want by a Cue of Preferences"
_ICLR.cc/2021/Conference — Reject_

### Official Review · AnonReviewer3 · 2020-10-26
**Interesting results, but lacks clarity in the presentation and the soundness of the approach needs more evidence**

**Rating:** 5
**Confidence:** 4

**Review:**

- Overview:
	The paper addresses the problem of training a GAN to match the distribution of part of the dataset called the 'desired data distribution', instead of the whole dataset as usually done in the context of GANs. This problem can be of interest when for instance the 'desired training data'  is limited and one would like to leverage additional 'undesired' data for training while still ensuring the model generates samples similar to the desired data.
	The address this problem, the authors propose to use an additional set S of pairwise samples  (X_1,X_2) called the set of preferences to guide the training of a WGAN. This set specifies that the user prefers sample X_1 over X_2.  X_1 is typically  a sample from the desired data and X_2 is an undesired sample. This provides an additional information during training to favor samples that are similar to X_1.
	The authors then propose to encourage the critic to give a higher score to sample X_1 and a lower score to sample X_2. This is achieved by adding a penalty to the WGAN loss. This  penalty is simply given by the margin loss between the scores of X_1 and X_2.
	The critic loss is then a tradeoff between two terms: the standard WGAN term which encourages 'data quality' and the preference term which favors samples similar to the preferred data.
	 The authors then identify two major training situations which determine how easily the tradeoff can be achieved: 'minor correction' and 'major correction'.
	- The minor correction is when the 'desired data' is very similar to the whole dataset. Thus the tradeoff is easily achieved using the proposed loss of eq 4.
	- The major correction is the harder situation where the 'desired' data are different from the whole dataset. Thus using the proposed method directly could either learn to preference at the expense of sample quality or maintain good quality but without learning the preference. To address this, the authors propose a modification of the algorithm that consist in replacing part of the training samples by generated ones at every epoch. The authors then claim that this addresses the problem since the modified training data should get closer to the desired data at every epoch.
	The authors then demonstrate empirically the ability of the proposed methods to successfully learn the desired distribution while using shared structure with the undesired data. This is first illustrated in a simple synthetic example in section 4. then in section 5, the author compare the proposed method with other baselines on MNIST and CelebA datasets.

	The authors conclude that the proposed method has two advantages compared to existing methods.
	- The method doesn't need a pre-defined score function to distinguish between desired and undesired data. That is because they instead rely on the pairwise preference set S.
	- While other methods do not use undesired data for training, the proposed method can effectively exploit those data to yield better performance when the desired dataset is scarce.

- Strength of the paper:
	The author show experimentally that the proposed method is able to take advantage of negative samples to learn the user-desired distribution. This allows to use fewer supervision than the method FBGAN which only relies on positive samples. This is illustrated in the experiment in 5.1.2. where the number of supervision is limited in purpose.

- Weakness of the paper:
	The main weakness is the soundness of the approach and the clarity of the paper.
	- In definition 1 (problem setting), the  authors first claim in the that proposed method allows to learn the desired distribution. But it remains unclear from the structure of the loss whether this is the case. It seems that the loss achieves only a tradeoff between learning the desired data and being close the whole training set. This is in particular apparent from the discussion about the minor and major correction scenarios. The statement in definition 1 should be adapted to reflect this limitation of the method and avoid confusion.
	- In the major correction scenario, the authors propose to introduce fresh generated samples as part of the training set at every epoch. By doing so, the authors claim that the modified training set is getting closer and closer to the desired distribution thus leading to a decreasing sequence of distances to the desired data. I honestly don't see how this solves the problem:
		- Why would one obtain a decreasing sequence as claimed?
		- How does it prevent the quality of the generated data from being degraded? It doesn't seem like using bad generated samples as part of the training set could help prevent the degradation of sample quality.
		- It would be good to have either theoretical or empirical evidence of why this approach is relevant.

	- The authors present a limitation of FBGAN as needing a predefined score function, while the proposed method relies on a set of pairwise preferences. Yet in practice, it seems that the set S is construction using a classifier or from the labels of the data. Isn't it equivalent to having  a pre-defined score function?

	- In 5.1.1, the authors compare the number "effective pairs" used during training in the case of FBGAN and the proposed method. For the proposed method, this is simply the number of elements in the set S. But for the competing method FBGAN which doesn't rely on pairs, I don't see what it means. The authors say that is corresponds to the product of desired and undesired samples. Is that a relevant quantity to consider for FBGAN? The authors then compare those numbers and conclude that the proposed method requires way fewer effective pairs. But I'm not convinced this is a fair comparison. Perhaps what could be more relevant is to simply compare the performance as a function of the number of epochs:  FID score, percentage of desired samples every epoch.

- Clarity:
	While the experiment sections are clear  overall, I found section 3 to be very confusing:
	- My current understanding is that, the score loss encourages the critic to take into account a preference of the user. Such preference is supposed to be encoded by the set of 'pairwise preferences' S. The procedure makes sense to me if the set $S$ is pre-determined by the user.
	- However, section 3.2.1 is confusing in that it suggests that the set S is constructed using the critic itself. More precisely, what was confusing is that the ranking between samples (which is used to construct the set S in equation 2) is defined by means of the score function $f$ (the last two sentences before eq 3.). Then the author say that the score $f$ is chosen to be the critic $D$ (which is learned during training). This clearly wouldn't makes sense, since no information about the user's preference is included.  Later in the experiments sections, the authors clarify how the set S is pre-defined, but still this makes the reading experience very confusing.
	- The background section do not contain relevant concepts and work that are key to understand the paper: The authors take inspiration from Relative GAN to designing the critic, but we don't know exactly how. This should be discussed in more details before presenting the method. The same is valid for FBGAN, a competing method, which is partially introduced in the method section. Only later, in the experiment section, that the reader discovers some subtleties in how FBGAN are trained. In particular, the authors explain in section 5 that the generated samples need also to be scored during training in the case of FBGAN, however, this is never mentioned before and seems to be critical to understand the experiments of section 5.1.2 (limited supervision scenario).

	 - In 5.1, the authors say that they use a pre-trained classifier to distinguish between desired and undesired generated data. This suggests that the generated samples are also used to construct the set S? However, isn't S an input to the algorithm and thus independent of the generator? Are those samples only used to evaluate to what extend the generator learns the preferred distribution?


- Questions and suggestions:
	- The training algorithm should be in the main paper.
	- How does the gap in performance between the two methods evolve as the number of supervision increases?
	- Is this behavior consistent across datasets? For instance on CelebA? It seems like on CelebA both methods perform equally well. However, showing that the method still requires fewer supervision in the case of datasets such as CelebA could be strengthen the results.
	- Why do the critic and generators need to be pre-trained? What happens when training from scratch?
	- 'As MNIST contains digits from 0 to 9, 0 is the most desirable data'  This sentence doesn't really make sense, or does it?

Updated review: Thank you  for your response and  for your efforts in addressing those points. I raised my score to 5 for the clarifications made to the document. However, I still think there are unjustified claims about the method, especially those related to the strong correction scenario and how introducing fresh generated samples is an effective procedure (conceptually / theoretically).

---

> ### Author Response · Authors · 2020-11-19
> **Concerns about the convergence of DiCGAN and the empirical comparison study (part 1/2)**
>
> Thank you very much for your comments.
>
> Weakness 1:	the concern about that definition 1 should reflect the limitation of the method.
>    - Response: The problem setting only clarifies the considered problem in this paper. It did not give claims for the proposed method.
>
> Weakness 2:	In the major correction scenario, 1) Why would one obtain a decreasing sequence as claimed? 2) How does it prevent the quality of the generated data from being degraded? It doesn't seem like using bad generated samples as part of the training set could help prevent the degradation of sample quality. 3) It would be good to have either theoretical or empirical evidence of why this approach is relevant.
>    - Response: (1) At every epoch, introducing the generated data into the training set will increase the ratio of desired data in the training set rather than degrading the quality of data. Therefore, DiCGAN converges as expected along the training process.
>
>    Specifically, referring to equation 4, the vanilla WGAN loss encourages the critic to assign high critic values for real data and low critic values for fake data. Meanwhile, the ranking loss encourages the critic to assign high critic values for the desired data and low critic values for the undesired data. Thus a sample with high critic value is not only desired but also high-quality. DiCGAN facilitates the generation of samples with high critic values, thus generating high-quality desired samples.
>
>    (2) Our DiCGAN will not shift the generation toward the bad quality direction as clarified in point (1). But introducing the generated samples into the training set to promote the ratio of the desired data may degrade the quality of training set since a perfect generated distribution is hard to obtain for complex datasets.  The same is true for FBGAN as FBGAN also introduces the generated samples into the training set. The problem can be eased up when GAN with extremely high-quality generation like BigGAN is used.
>
>    (3) In the appendix (originally submitted as the supplementary material), we have already shown empirical evidence that the ratio of desired data increases with the training epoch.
>
> Weakness 3:	The authors present a limitation of FBGAN as needing a predefined score function, while the proposed method relies on a set of pairwise preferences. Yet in practice, it seems that the set S is constructed using a classifier or from the labels of the data. Isn't it equivalent to having a pre-defined score function?
>    - Response: The reason we construct $S$ from the labels of the data is for fair comparisons with other baselines, which use explicit labels to derive the desired data distribution. However, our method can also be applied when the pairwise preferences are obtained from the user, whereas other baselines cannot.
>
> Weakness 4: 1) the concerns about the calculation of "\#effective pairs" for FBGAN and a fair comparison; 2) the suggested relevant comparison that compares the performance as a function of the number of epochs: FID score, percentage of desired samples every epoch.
>    - Response:  "\#effective pairs" is defined as the number of pairwise comparisons implied in the various ordering supervision.
> When FBGAN selects desired data, it is equivalent to pick up the top $K$ samples out of the whole dataset ($K$ is the number of desired samples). $K*(N-K)$ is the minimal pairs that verify which are top $K$ samples out of the whole dataset. That is why we use the product of desired and undesired samples to represent the number of effective pairs used in FBGAN.
> "\#effective pairs" compares the supervision amount of different methods, which reflects the extent of extra knowledge required by different methods.
>
>    The current calculation for FBGAN is the minimal number of used effective pairs to select top $K$ samples. The calculation for our DiCGAN is the number of actually used effective pairs. In fact, it is a little unfair for our method due to a possible underestimation of effective pairs for FBGAN.
>
>    The comparison of the percentage of desired samples versus every epoch, which compares the performance of various methods.
>
>    These two metrics evaluate the performance of methods from different aspects.
>
>    For a more thorough comparison, we also compared the percentage of desired samples versus every epoch, added in the appendix.

---

> > ### Author Response · Authors · 2020-11-19
> > **Reply to the concerns about the clarity and questions (part 2/2)**
> >
> > Clarity 1: confusion about the construction of pairwise preference set $S$.
> >    - Response: Sorry for confusing the reviewer. The set $S$ is indeed pre-determined by the user. The score-based ranking model learns a score function from the pairwise preferences $S$, where the ranking scores can indicate the user preference. That is, $x_1 \succ x_2$, then $f(x_1)-f(x_2)>0$. Our critic would learn from $S$ like the ranking model.
> >
> > Clarity 2: concern about the introduction of RGAN and FBGAN.
> >    - Response: RGAN is a variant of GAN that also targets at learning the whole data distribution. It takes the critic value to describe the sample quality and uses a pairwise ranking loss like objective that aims to assign higher critic values for real samples.
> >
> >    FBGAN targets at learning the user-desired distribution by iteratively introducing desired samples into the training data. Specifically, FBGAN is pretrained using the vanilla WGAN. In each training epoch, the generator first generates certain amounts of samples. The generated samples possessing the desired properties are selected by an expert selector and used to replace the old training data. Then, regular WGAN is trained with the updated training data. Since the ratio of the desired samples gradually increases in the training data, all training data will be replaced with the desired samples. Finally, FBGAN
> > would derive the desired distribution when convergence.
> >
> >    We introduce RGAN and FBGAN more clearly in the Introduction of the revised paper.
> >
> >  Clarity 3: concern about the construction of $S$ from the generated samples.
> >    - Response: In terms of the minor correction situation, the pairwise preferences are only constructed from the real training data. In terms of the major correction situation, as we introduce generated samples into the training dataset, the pairwise preferences will be constructed from the introduced generated samples also.
> >
> > Questions: 1) The training algorithm should be in the main paper. 2)How does the gap in performance between the two methods evolve as the number of supervision increases? 3)Is this behavior consistent across datasets? For instance on CelebA? It seems like on CelebA both methods perform equally well. However, showing that the method still requires fewer supervision in the case of datasets such as CelebA could strengthen the results. 4)Why do the critic and generators need to be pre-trained? What happens when training from scratch? 5) 'As MNIST contains digits from 0 to 9, 0 is the most desirable data' This sentence doesn't really make sense, or does it?
> >
> >    - Response: 1. Thanks for the suggestion. We have already put the training algorithm in the main paper.
> >
> >    2) We added the experiments to clarify the performance gap between the two methods versus the number of supervision in the appendix.
> >
> >    3) The suggested experiments are running. We will report it as soon as possible.
> >
> >    4) Pretraining the model would make the convergence faster. Note that it is equivalent to set up $\lambda=0$ at the first stage of training.
> >
> >    5) We revised it to 'Zero is the smallest digit in MNIST, thus as the desired data.'

---

### Official Review · AnonReviewer2 · 2020-10-27
**GAN + Score based ranking preferences**

**Rating:** 5
**Confidence:** 4

**Review:**

**Overview**
The paper proposed differential critic GAN, which proposes a differential critic that learns the preference direction from pairwise preferences. A corresponding loss function is added to the WGAN loss to ensure that the model learns samples that have high rank. Empirical results show that the method is able to reflect the ranking from the user.

**Strengths**
- The paper presents a decent solution to the problem where additional rank-based supervision is provided.
- The paper have superior performance than FBGAN and WGAN on specific settings concerning MNIST and CelebA.

**Weaknesses and Questions**
- Why would we consider pairwise preferences over explicitly labeling what the users consider to be good data or not?
- Do the preferences define a partial order? Can user preferences be cyclic?
- The problem statement seems to be flawed; when we say x1 ranks higher than x2, does that mean we want x1 to appear in the distribution and x2 to not appear in distribution (I suppose not)? If the answer is no, then of what percentile (assuming total order, that is) do we consider to have low probability (rank does not indicate weights which is eventually what we want from a well defined user-based data distribution)?
- Evaluation is a bit limited, especially in terms of sample quality. CelebA-HQ inception score does not count, because IS is not a good metric for image quality on human faces (with a untrained model you might get 3 and it gets lower as you train). The CelebA image quality is very far from that trained with a decent GAN (like smaller BigGANs, which you can train with 1 GPU in about a day).
- Why not use a CWGAN like objective where you only train on datapoints with c = 1? If your goal is to make use of the larger dataset for better generation, then there are some other work, then there are some other importance-weighted work to your interest: Choi et al. 2020, Fair Generative Modeling via Weak Supervision.

---

> ### Author Response · Authors · 2020-11-19
> **Our DiCGAN always learns the desired distribution even given limited supervision while ensuring the image quality as good as possible, whereas the state-of-art FBGAN fails.**
>
> Thank you very much for your comments.
>
> Comment 1:	Why would we consider pairwise preferences over explicitly labeling what the users consider to be good data or not?
>    - Response: This is for fair comparisons with other baselines, which use explicit labels to derive the desired data distribution.
>
>    Note that there may not exist a universal criteria for users to label whether the sample is desired or not. Even the criteria exists, it may have to access the whole dataset in order to select top K samples as good ones, which actually utilizes much more supervision information compared to our method, supported by the comparison study w.r.t. the number of effective pairs.
>
>    Our method can be directly used when true pairwise preferences annotated by the users are given but other methods cannot.
>
> Comment 2:	Do the preferences define a partial order? Can user preferences be cyclic?
>    - Response: Yes, the pairwise preferences define a partial order. The partial ranking can derive a total ranking when the pairwise preferences are complete according to the learning to rank theory [1].
>
>    Yes, we can apply noise preference ranking techniques to our model so as to handle the cyclic user preferences.
>
> Comment 3:	1) When we say x1 ranks higher than x2, does that mean we want x1 to appear in the distribution and x2 to not appear in the distribution? 2) Of what percentile (assuming total order, that is) do we consider to have low probability (rank does not indicate weights which is eventually what we want from a well-defined user-based data distribution)?
>    - Response: When x1 ranks higher than x2, it means that the user prefers x1 over x2 to appear in the learned distribution.
>
>    The distribution of the top-ranked data is desired, which is automatically learned from the pairwise preference by the model.
>
> Comment 4:	1) Evaluation is a bit limited, especially in terms of sample quality. 2) The CelebA image quality is very far from that trained with a decent GAN (like smaller BigGANs, which you can train with 1 GPU in about a day).
>    - Response: We will use more evaluation metrics to evaluate the quality of generated face images.
>
>    It is possible to use a more advanced GAN to improve quality.
>
>    But please note that our main target is to learn the desired data distribution while ensuring the image quality as good as possible. Our method can always learn the desired distribution even given limited supervision, whereas the state-of-art FBGAN fails.
>
> Comment 5:	Why not use a CWGAN like objective where you only train on datapoints with c = 1? If your goal is to make use of the larger dataset for better generation, then there are some other works, then there are some other importance-weighted works to your interest: Choi et al. 2020, Fair Generative Modeling via Weak Supervision.
>    - Response: Using a CWGAN like objective training on datapoints with c=1 will degenerate to one of our baselines WGAN. We also consider CWGAN as baseline to learn conditional distribution $p(x|c=1)$ as desired distribution.
> But their performances are both limited, especially given the limited desired data.
>
> Our goal is to learn the distribution of user-desired data when only partial instead of the entire dataset possesses the desired properties. The suggested work is not related.
>
> Reference
>
>    * [1] Lu, Tyler, and Craig Boutilier. "Learning Mallows models with pairwise preferences." ICML. 2011.

---

> > ### Comment · AnonReviewer2 · 2020-11-19
> > **I still don't get the "distribution of the top-ranked data" bit**
> >
> > If my generator is perfect, what should it generate? What is the equilibrium?
> >
> > It seems to me that generator is always incentivized to produce only the top sample. Here is why:
> > - Let's suppose that discriminator assigns scores such that the rank k is always m larger than the rank k-1 one. So rank loss is zero. Let's say there are n values, so score of top sample is mn.
> > - The generator achieves the highest value it can get, which is the top value mn.
> > - The discriminator cannot decrease the score for the top value here, because if it did, it also affects the rank-based score. So if discriminator makes the top score (mn-1), generator score decreases by 1, but discriminator also loses a score of 1 from the ranking (assume that m > 1).
> >
> > So this seem to suggest that the state above is a (locally neutral) equilibrium to your minimax problem.

---

> > > ### Author Response · Authors · 2020-11-19
> > > **The equilibrium of DiCGAN**
> > >
> > >
> > > If the generator is perfect, it would only generate the desired samples. The equilibrium is that the generated distribution is equal to the desired data distribution.
> > >
> > > We first clarify some missing points.
> > > There are two types of pairwise preferences, pairs with significant preferences, and comparable pairs.
> > >
> > > In our model, we only considered pairs with significant preferences,  but simply ignored pairwise preferences that are comparable. Since the learned critic (a ranking model) did not discriminate the difference between comparable pairs, the resultant top-ranked data would be a collection of samples that are comparable.  It means that the number of significant preferences would decrease when we introduce the generated samples (the top-ranked data) into the training data at each epoch.
> > >
> > > We explain the equilibrium of DiCGAN from the minor correction and the major correction.
> > >
> > >    * In terms of the minor correction, by assigning a proper $\lambda$, a proper preference direction is enforced on the critic. Then the generated data is guided towards the desired data.
> > >
> > >    * In terms of the major correction, a large correction may not ensure the quality of the generated data, since the WGAN loss, used to guarantee the image quality, is defined between the generated data and the whole real data. Therefore, we split the correction into a sequence of minor corrections. At every minor correction, a relatively small $\lambda$ is set to enforce the critic to be slightly biased to the desired data direction. Thus the generated data can be guided slightly biased to the desired data while is not far away from the training data. Then the generated data will be introduced into the training data to ensure that the ratio of desired data increases at each epoch.  Accordingly, the number of significant pairs decreases along with the training. The ranking loss decreases as well since more comparable pairs will be ignored by the ranking loss. When converging, all training samples are desired. There are no significant preferences in the training data. The ranking loss would be zero at this time. DiCGAN degenerates to the original GAN.

---

> > > > ### Comment · AnonReviewer2 · 2020-11-19
> > > > **What are pairwise preferences that are comparable?**
> > > >
> > > > As you said, the pairwise preferences define a partial order. Then does your objective perform as desired, if I already knew all the pairwise preferences and use all of them in the training objective? Or does the generator produce all maximal ranking samples only?
> > > >
> > > > If this is the case, then it means that we should just remove the data as long as it is less preferred by some other data, since the samples will not be in the generator for the equilibrium anyways?

---

> > > > > ### Author Response · Authors · 2020-11-20
> > > > > **The objective is optimized to shift the generated data gradually towards the desired data since the whole object is not directed by the ranking loss only.**
> > > > >
> > > > > Take learning the distribution of small MNIST digit as an example. Users are asked to determine which of a pair of MNIST digits is “smaller’’.  The user would say “digit zero is smaller than digit one”. Rather, given two zero digit images, the user would say “they are comparable”.
> > > > >
> > > > > Note that there is a balance between the vanilla WGAN loss and the ranking loss, which is controlled by $\lambda$.  The objective is optimized to shift the generated data gradually towards the desired data rather than make the generator produce all maximal ranking samples once for all since the whole object is not directed by the ranking loss only.
> > > > >
> > > > > In terms of the minor correction where the majority of data are desired data, optimizing the above two objectives will automatically remove the small amount of data that is less desired compared to the majority data.
> > > > >
> > > > > However, in terms of the major correction where the minority of data are desired data, we cannot simply remove the undesired data that is the majority, which would degrade the image quality simultaneously. It is because the WGAN loss, used to guarantee the image quality, is defined between the generated data and the whole real data. Therefore, we split the major correction into a sequence of minor corrections so as to satisfy the small discrepancy constraint between the current generated distribution and the current training data distribution. Such a strategy can maintain image quality as good as possible.

---

> > > > > > ### Comment · AnonReviewer2 · 2020-11-20
> > > > > > **In the MNIST example, should I produce any images that are labeled 1?**
> > > > > >
> > > > > > I am not too concerned about image quality at all. I assume that you have infinite data and infinite modeling capacity. I am just a bit worried that is unclear what is the solution that we want to achieve here. It seems that the goal here is still to produce the highest ranking data (if $\lambda \geq 1$); if $\lambda < 1$, then what would the model do precisely if $m = 1$?
> > > > > >
> > > > > > It also seems that you want to leverage low rank data to achieve high image quality, and that is their only use under this model. If this is true, then this point does not seem to be conveyed in the paper very well.

---

> > > > > > > ### Author Response · Authors · 2020-11-21
> > > > > > > **In the MNIST example,  the generator is to produce images that are labeled zero.**
> > > > > > >
> > > > > > > In the MNIST example, the convergence state of the generator is to only produce images that are labeled zero as the digit zero is the smallest digit in MNIST that contains digit zero to nine (See Fig. 3).
> > > > > > >
> > > > > > > When $\lambda$ is large, the ranking loss dominates the training of DiCGAN. The generator will produce the highest-ranking data. However, the generated data may not be desired as expected. This is because the pairwise preferences are constructed only on the training data, which cannot generalize to the data that is out of the training data distribution. Therefore, the ranking score on the generated data will lose its physical meaning (i.e., indicating desired or undesired) when the generated data distribution exhibits a large discrepancy from the training data distribution. Namely, the generated data may be some meaningless images but with the highest-ranking score.
> > > > > > >
> > > > > > > When $\lambda$ is small, the ranking loss enforces a small shift on the generated distribution toward the desired data distribution. Through an iterative paradigm, the generated distribution converges as desired.
> > > > > > >
> > > > > > > $m$ is the ranking margin that denotes the difference of ranking scores between two samples in terms of the degree of user preference. It is not a focus in this work and we simply fix it to $1$.
> > > > > > >
> > > > > > > The function of low-rank data is twofold: (1) ensuring high image quality; (2) providing the preference direction together with the high-rank data. We will clarify this in the new version.

---

### Official Review · AnonReviewer1 · 2020-10-29
**Extending GAN with pairwise loss/regularization on the discriminator**

**Rating:** 5
**Confidence:** 4

**Review:**

The motivation of this study is to estimate the distribution of desired data from the entire data distribution. And the proposed solution extends existing GAN solutions by introducing an additional pairwise loss on the discriminator, e.g., its scores on the desired instances should be higher than the undesired ones. The idea is natural and neat, and it is also proved to be effective in the reported experiments.

First of all, why not add such pairwise regularization to the generator? I did not find any discussion on this alternative and the advantage of adding regularization on the discriminator side. At least, it is important and interesting to empirically compare these two choices to understand their difference. On a related note, as the proposed DiCGAN is very similar to RGAN, it is important to compare these two solutions to better understand how their design difference translates to empirical performance difference.

I do not fully follow the discussion in Section 3.3, especially the part about the major correction aspect. The claim is when the desired data distribution is far away from the entire data distribution, the generator might have difficulty in satisfying the WGAN loss. And then the proposed solution is to gradually generate new instances to replace the old one. But I am not sure why this would lead to convergence, although empirical results provided in Appendix shows this strategy. For example, in the first round, the quality of generator is bad, such that the pairs constructed from the generated instances do not reflect the preference direction. Hence the only pairs useful there are those from the initial training instances, which might still require major correction on the generator? And I also did not find any discussion about how the number of pairwise samples or the way to sample such pairs affects the model’s performance. As the pairs are constructed from individual instances, transitivity among the pairs would introduce additional complexity in model training, as the pairs are no longer independent.

**Acknowledgement of author responses**
The authors' responses were helpful to clarify the settings and basic ideas of the proposed solution. I highly appreciate their effort. However, the limitations of the proposed solution and limited novelty still outweigh the merit; and therefore, I will keep my original evaluation.

---

> ### Author Response · Authors · 2020-11-19
> **Concerns about the pairwise regularization and the convergence of the method**
>
> Thank you very much for your comments.
>
> Comment 1:	Why not add such pairwise regularization to the generator? I did not find any discussion on this alternative and the advantage of adding regularization on the discriminator side. At least, it is important and interesting to empirically compare these two choices to understand their difference.
>    - Response: Thanks for this valuable suggestion. Yes, it is possible to consider a pairwise regularization to the generator.
>
>    As the target is to learn the desired distribution, the regularization to the generator can be making the score for the generated samples larger than the undesired samples. We construct the regularization with a principle similar to FBGAN. Specifically, a selector is first applied to give a full ranking for the training data, and then bottom $K$ samples are picked up as the undesired samples. The pairwise preferences are then defined over the generated samples and the undesired samples.
>
>    We now consider the above-mentioned method as another baseline. We have added such discussion in the revision and is working on the experiments. The experimental results will be reported as soon as possible.
>
> Comment 2:	As the proposed DiCGAN is very similar to RGAN, it is important to compare these two solutions to better understand how their design difference translates to empirical performance difference.
>    - Response: RGAN is not proposed to learn the desired data distribution but to learn the whole data distribution as the vanilla GAN. Our proposed DiCGAN just shares a similar understanding of the critic value as RGAN. We added more discussion in the Introduction of the revised paper to avoid confusion.
>
> Comment 3 point 1: I do not fully follow the discussion in Section 3.3, especially the part about the major correction aspect. The claim is when the desired data distribution is far away from the entire data distribution, the generator might have difficulty in satisfying the WGAN loss. And then the proposed solution is to gradually generate new instances to replace the old one. But I am not sure why this would lead to convergence.
>    - Response: At every epoch, introducing the generated data into the training set will increase the ratio of desired data in the training set rather than degrading the quality of data. Therefore, DiCGAN converges as expected along the training process.
>
>    Specifically, referring to equation 4, the vanilla WGAN loss encourages the critic to assign high critic values for real data and low critic values for fake data. Meanwhile, the ranking loss encourages high critic values to be assigned to the desired data while low critic values are assigned to the undesired data. **Thus a sample with high critic value is not only desired but also high-quality.** DiCGAN facilitates the generation of samples with high critic values, thus generating high-quality desired samples.
>
> Comment 3 point 2: I also did not find any discussion about how the number of pairwise samples or the way to sample such pairs affects the model’s performance.
>    - Thanks for your suggestion. We explore how the model performance is affected with regards to the number of pairwise samples using MNIST dataset. The number of used pairwise preferences is set to $5K,25K,50K,250K$, respectively. It shows that our DiCGAN is not sensitive to the number of pairwise preferences, since DiCGAN can always capture the small digit distribution under all settings. We include the study in the appendix.
>
>    In terms of the way to sample pairs, as we do not have any prior knowledge for the data, random sampling pairs from the training data is the most unbiased one, which is adopted in this work.
>
> Comment 3 point 3: As the pairs are constructed from individual instances, transitivity among the pairs would introduce additional complexity in model training, as the pairs are no longer independent.
>    - Response: Yes, we agree that training with pairs introduces more additional complexity in model training than training with individual instances. But regarding the target of this paper, i.e., learning the desired data distribution, pairwise preferences are simple and easily accessible supervision that reflect user preference. On the contrary, the labels that indicate whether the individual instances are desired or not may not exist due to a lack of universal user criteria.
>
>    Meanwhile, we tailor-designed the differential critic to consider two goals -- image quality and preference direction-- at the same time, which marginally increases GAN's complexity.

---

### Official Review · AnonReviewer4 · 2020-10-30
**Review for "Differential-Critic GAN: Generating What You Want by a Cue of Preferences"**

**Rating:** 5
**Confidence:** 4

**Review:**

The authors introduce DiCGAN, an algorithm to learn a generative model that comes up with samples whose likelihood is based on a real dataset but adjusted given user preferences. They train the critic to assign high values to samples with higher preference values and thus the generator tends to move its samples towards these points. The idea is nice and reasonably novel in my opinion, but the paper has quite a few problems.

The first problem is that the writing of the paper is awful. Already in the abstract there are several syntactical and semantical errors. It is actually fairly hard to read this paper, likely the idea is simple and one can understand it from the equations, but everything that's written pretty much makes the paper harder to read. The authors also write too strong claims for a scientific paper: many times they write that DiCGAN learns the user desired data distribution (learning a high dimensional data distribution with finite data is not a possible not an interesting goal, the users should use "approximates" instead of "learns" and describe *how* it approximates it and what is lost), and things like "the superiority of DiCGAN is twofold". There's an entire section titled "Superiority of DiCGAN over FBGAN"!
The paper is also quite imprecise / uses the wrong mathematical terms at points. For instance, "equation 4 is the Lagrange function of equation 6".

Finally, while the experiments are interesting, they're all on MNIST or aligned celeb-A in 64x64, and the samples are terrible. It is hard to believe that this method could be scaled up as is, or at least there is little evidence to that regard.

---

> ### Author Response · Authors · 2020-11-19
> **The paper is proofread to avoid syntactical errors.**
>
> Thank you very much for your comments.
>
> Comment 1:	Concerns about the writing of the paper and the value of the considered research problem.
>    - Response: The way we write this paper is for better analyzing our method, especially for analyzing the minor correction situation and the major correction situation.
>
>    In GAN-related literature [1,2,3], "approximating" a distribution is widely described as "learning" the distribution". That is why we denote "learning the desired data distribution".
>
>    We disagree with the reviewer that "learning a high dimensional data distribution with finite data is not possible and not an interesting goal". We admit that such a goal is difficult but not impossible. Actually, this direction recently attracts considerable attentions [4,5]. Meanwhile, there is one paper [6] in NeurIPS2020 considering a similar problem as ours. They also consider an application of learning an under-represented class in an unbalanced dataset, which is exactly the same as one of our experimental setting. Last but not least, our task of learning the desired data distribution has many potential applications, such as a Nature Machine Intelligence paper [7] considered, protein designs in the synthetic biology area.
>
>    We have proofread our paper to avoid syntactical and semantic errors.
>
>    Accordingly, we change our claim as "the advantages of DiCGAN are twofold", "Comparison of DiCGAN and FBGAN". We revised our sentence to "equation 4 is the Lagrangian function."
>
> Comment 2:	While the experiments are interesting, they're all on MNIST or aligned celeb-A in 64x64, and the samples are terrible. It is hard to believe that this method could be scaled up as is, or at least there is little evidence to that regard.
>    - Response: We have done experiments on CIFAR. But due to the limited space, we put their results on CIFAR dataset in the appendix (originally submitted as supplementary materials).
>
>    The main goal of this paper is to learn the desired data distribution. Our method can learn the desired data distribution even when the supervision is limited, while the state-of-art method FBGAN would fail under such a scenario. The quality of the samples generated by our DiCGAN is relatively comparable to that of FBGAN.
>
> Reference
>
>    * [1] Arjovsky, M., Chintala, S., \& Bottou, L., Wasserstein generative adversarial networks, ICML, 2017.
>
>    * [2] Kurach, Karol, et al., "A large-scale study on regularization and normalization in GANs." ICML, 2019.
>
>    * [3] Arora, Sanjeev, Andrej Risteski, and Yi Zhang. "Do GANs learn the distribution? some theory and empirics." ICLR. 2018.
>
>    * [4] Karras, T., Aittala, M., Hellsten, J., Laine, S., Lehtinen, J., \& Aila, T., Training generative adversarial networks with limited data, NeurIPS, 2020.
>
>    * [5] Noguchi, A., \& Harada, T., Image generation from small datasets via batch statistics adaptation, ICCV, 2019.
>
>    * [6] Asokan, Siddarth, and Chandra Seelamantula. "Teaching a GAN What Not to Learn." NeurIPS, 2020
>
>    * [7] Gupta, A.,\& Zou, J., Feedback GAN (FBGAN) for DNA: a novel feedback-loop architecture for optimizing protein functions, Nature Machine Intelligence, 2019.

---

### Author Response · Authors · 2020-11-19
**To all reviewers: the main contribution of our work**

The main goal in this work is to learn the distribution of user-desired data when only partial instead of the entire dataset possesses the desired properties while ensuring the image quality as good as possible.

The main contribution of our DiCGAN can be summarized as:
1) DiCGAN always learns the desired distribution even with limited supervision, whereas previous methods fail.
2) Previous methods rely on user-defined criteria, which may not exist in real applications. DiCGAN resorts to simple and accessible pairwise preferences. It is promising in many real-world applications. Such as in the Recommendation system. Clicks-through data can be easily collected as pairwise preferences. DiCGAN can be applied to generate what the user expects from its preferences.
3) The tailor-design differential critic considers two goals -- image quality and preference direction simultaneously, which marginally increases GAN's complexity.

**We updated a new version of the paper. We highlighted the main revision in red.**

---

### Decision · Program_Chairs · 2021-01-07
**Final Decision**

**Decision:**

Reject

**Comment:**

The paper presents a method to regularize the discriminator in  GAN training with a ranking loss based on the user preference for a desired set within a larger dataset. The tradeoff between GAN loss and preference loss dependence on the distance of the set to the full dataset and the authors consider two regimes : "small and major correction". A major correction is needed when the targeted set is very different from the whole density, authors propose in this scenario to replace samples from the data by samples from the generator. The setting in the paper is interesting and can be useful in practice.

There was a lengthy discussions between the authors and the reviewers, the discussion pinpointed issues , some of them were addressed in the rebuttal . Some issues remain unanswered regarding the clarity and some claims in the paper.

The clarity of the paper needs further improvement and  1)  clarify section 3  the setup and the background section   2)  justify claims about the method, in  the strong correction scenario  when fresh generated samples are introduced how  is this an effective procedure? (conceptually / theoretically).